# MagicRemover: Tuning-free Text-guided Image Inpainting with Diffusion Models

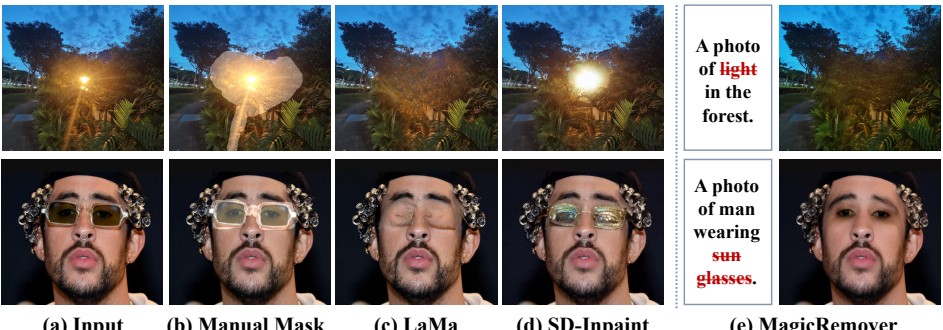

**(a) Input**  **(b) Manual Mask**  **(c) LaMa**  **(d) SD-Inpaint**  **(e) MagicRemover**

Figure 1: **MagicRemover:** We present a tuning-free image inpainting approach that requires only a textual input from users, rather than manual binary masks, to erase the desired objects. From left to right are: (a) input image, (b) manual mask, mask-based inpainters (c) LaMa (Suvorov et al., 2021) and (d) SD-Inpaint (Rombach et al., 2022b), and (e) our MagicRemover. Our method is capable of eliminating areas that are hard to be defined with binary masks.

## Abstract

Image inpainting aims to fill in the missing pixels with visually coherent and semantically plausible content. Despite the great progress brought from deep generative models, this task still suffers from *i.* the difficulties in large-scale realistic data collection and costly model training; and *ii.* the intrinsic limitations in the traditionally user-defined binary masks on objects with unclear boundaries or transparent texture. In this paper, we propose **MagicRemover**, a tuning-free method that leverages the powerful diffusion models for text-guided image inpainting. We introduce an attention guidance strategy to constrain the sampling process of diffusion models, enabling the erasing of instructed areas and the restoration of occluded content. We further propose a classifier optimization algorithm to facilitate the denoising stability within less sampling steps. Extensive comparisons are conducted among our MagicRemover and state-of-the-art methods including quantitative evaluation and user study, demonstrating the significant improvement of MagicRemover on high-quality image inpainting.

## 1 Introduction

Image inpainting is the task of filling in the missing or corrupted regions of an image, which shows important implications in many applications like image editing, object removal, image restoration, to name a few. The key challenge in image inpainting is to make the filled pixels consistent with the non-missing areas to form a photo-realistic scene. Most existing state-of-the-art methods (Guo et al., 2021; Suvorov et al., 2021; Li et al., 2022; Zeng et al., 2021) are developed upon Generative Adversarial Networks (GANs) (Goodfellow et al., 2020) and show impressive capacity on generating visually plausible and consistent content in image holes with arbitrary sizes and shapes. For model training, a large amount of corrupted-painted image pairs are synthesized by applying a set of predefined irregular binary masks (Suvorov et al., 2021) on the original images. Although the large-scale training samples can facilitate the generalization of inpainting models, the synthetic corrupted regions during training still have large gap with the requirement of real-world users, leading to a

limited performance on the removal or completion of diverse objects. On one hand, a high-quality user-defined mask is always preferred to reduce models' sensitivity for more robust inpainting results, which inevitably brings inconvenience and is especially hard to draw on the objects with complex structures or unclear boundaries (see the first row of Fig. 1). On the other hand, a binary mask with hard labels is not suitable to indicate semi-transparent areas like glass or reflection where background pixels are usually involved. In a binary mask, the transparent pixels and the inside background are treated equally for inpainting, easily incurring original background regions being destroyed (See the second row in Fig. 1). In such cases, an alpha matte (Wang & Adelson, 1993) with soft labels is demanded to separate the semi-transparent layers from RGB background, which however is impossible for users to label on real images.

Recently, diffusion models (Ho et al., 2020a) have emerged as the mainstream frameworks in various generative tasks (Rombach et al., 2022b; Yang et al., 2023; Singer et al., 2022). Trained on billions of text-image data (Schuhmann et al., 2022), Text-to-Image (T2I) diffusion models are capable of understanding text-vision correspondence and produce high-quality, diverse and realistic image content. How to adapt the powerful diffusion in downstream tasks has become a hot research issue in the literature. One of the popular solutions (Yildirim et al., 2023; Haque et al., 2023; Brooks et al., 2023) is to leverage large-scale models (Brown et al., 2020) to synthesize large amounts of paired data, which are subsequently used to implement a standard fully training pipeline. However, such data synthesis based pipeline might not adapt well in image inpainting since *i.* the process of data synthesis and training is costly and relies on large-scale computational resources; *ii.* it's hard to find an inpainting method with robust generalization and promising performance on diverse distributions. In contrast to the heavy training framework, another research trend (Epstein et al., 2023; Hertz et al., 2022; Shi et al., 2023) is to explore an editable space in the internal representations of diffusion to allow image editing from various prompts like text or drag, without applying additional models, data or training. These methods (Hertz et al., 2022) found that the cross-attention shows great zero-shot capacity on highlighting the correspondence areas for textual tokens. It is then natural to ask whether we can leverage powerful generalization capacity of the pretrained diffusion to develop a zero-shot inpainting algorithm for natural images.

In this paper, we propose ***MagicRemover***, a tuning-free framework for text-guided inpainting, *i.e.,* erasing object following textual instructions. Our method leverage the pretrained diffusion model to construct an effective guidance from the internal attentions to constrain the process of object removal and occluded region restoration. Given the input image and the corresponding text instruction of interested objects, we first exploit reverse algorithm to project them into latent space, where the intra-domain self-attention and cross-domain cross-attention are calculated. Then, a novel guidance strategy is proposed and implemented on the attention maps to achieve soft object removal and background inpainting. Besides, we propose a classifier optimization algorithm to reduce the inversion steps while enhancing the editing capability and stability.

Our main contributions are summarized as follows:

1. We introduce *MagicRemover*, which takes advantage of the internal attention of pretrained T2I diffusion models to conduct text-guided object removal on natural images without auxiliary model finetuning or supervision.

2. We propose an attention-based guidance strategy together with a classifier optimization algorithm to enhance the captivity and stability of inpainting within less diffusion sampling steps.

3. We conduct sufficient experiments including quantitative evaluation and user study, demonstrating the superior performance of the proposed MagicRemover on object removal with soft boundaries or semi-transparent texture.

## 2 RELATED WORK

### 2.1 DIFFUSION MODELS

Recently, Diffusion Probabilistic Models (DPMs) (Ho et al., 2020a; Song et al., 2020b) have received increasing attention due to their impressive ability in image generation. Diffusion models have broken the long-term domination of GANs (Goodfellow et al., 2020) and become the new

state-of-the-art protocol in many computer vision tasks. DPMs consist of a forward and a reverse process. Given a sample from data distribution $\mathbf{x}_0 \sim p_{data}(\mathbf{x})$, the forward process gradually destroys the structure in data by adding Gaussian noise to $x_0$ for $T$ steps, according to the variance schedule $\beta_1, ..., \beta_T$:

$$q\left(\mathbf{x}_t \mid \mathbf{x}_{t-1}\right) = \mathcal{N}\left(\mathbf{x}_t; \sqrt{1 - \beta_t}\mathbf{x}_{t-1}, \beta_t\mathbf{I}\right). \tag{1}$$

Subsequently, a U-Net $\epsilon_\theta\left(\mathbf{x}_t; t\right)$ (Ronneberger et al., 2015) can be trained to predict the noise $\epsilon_t$ added during the forward process based on the given condition $\mathbf{y}$ with loss:

$$L(\theta) = \mathbb{E}_{t \sim \mathcal{U}(1,T), \epsilon_t \sim \mathcal{N}(0,\mathbf{I})}\left[w(t)\left\|\epsilon_t - \epsilon_\theta\left(\mathbf{x}_t; t, \mathbf{y}\right)\right\|^2\right], \tag{2}$$

where the condition $\mathbf{y}$ can be $\varnothing$, text (Ramesh et al., 2022) or images (Ho et al., 2022), etc. After the training is completed, the reverse process can trace back from the isotropic Gaussian noise $\mathbf{x}_T \sim \mathcal{N}\left(\mathbf{x}_T; \mathbf{0}, \mathbf{I}\right)$ to the initial data distribution using the predicted noise.

In order to control the randomness of the DDPM (Ho et al., 2020b) sampling process, DDIM (Song et al., 2020a) modifies the reverse process into the following form:

$$\mathbf{x}_{t-1} = \sqrt{\alpha_{t-1}}\underbrace{\left(\frac{\mathbf{x}_t - \sqrt{1 - \alpha_t}\epsilon_\theta\left(\mathbf{x}_t\right)}{\sqrt{\alpha_t}}\right)}_{\text{"predicted } \mathbf{x}_0\text{"}} + \underbrace{\sqrt{1 - \alpha_{t-1} - \sigma_t^2} \cdot \epsilon_\theta\left(\mathbf{x}_t\right)}_{\text{"direction pointing to } \mathbf{x}_t\text{"}} + \underbrace{\sigma_t\epsilon_t}_{\text{random noise}}, \tag{3}$$

where $\alpha_t = 1 - \beta_t$. By manipulating the value of $\sigma_t \in [0, 1]$, the randomness of the sampling process can be controlled. When $\sigma_t = 0$, the sampling process becomes a deterministic process.

Due to the extensive computational resources required for diffusion and denoising directly at the image level, Rombach et al. (2022a) introduces latent diffusion model. Latent diffusion model employs an encoder to map images to a latent space $\mathbf{z}_0 = \mathcal{E}(\mathbf{x}_0)$, then utilizes a decoder to map them back to the image space $\mathcal{D}(\mathbf{z}_0) \approx I$, while training a diffusion model in the latent space.

## 2.2 CLASSIFIER GUIDANCE

From the perspective of score matching (Vincent, 2011), the unconditional neural network $\epsilon_\theta\left(\mathbf{z}_t; t\right)$ optimized by the diffusion models is used to predict the score function $\nabla_{\mathbf{z}_t} \log p\left(\mathbf{z}_t\right)$. In order to more explicitly control the amount of weight the model gives to the conditioning information, (Dhariwal & Nichol, 2021) proposed classifier guidance, where the diffusion score $\epsilon_\theta\left(\mathbf{z}_t; t\right)$ is modified to include the gradient of the log likelihood of an auxiliary classifier model $p_c\left(\mathbf{y} \mid \mathbf{z}_t\right)$ as follows:

$$\hat{\epsilon}_t = \epsilon_\theta\left(\mathbf{z}_t; t\right) - s\sigma_t\nabla_{\mathbf{z}_t} \log p_c\left(\mathbf{y} \mid \mathbf{z}_t\right), \tag{4}$$

where $s$ is employed to modulate the gradient magnitude of the noisy classifier, thereby adjusting the degree to which the model is encouraged or discouraged from considering the conditioning information. The classifier model can also be replaced with a similarity score from a CLIP (Radford et al., 2021) model for text-to-image generation (Nichol et al., 2021), or an arbitrary time-independent energy as in universal guidance (Bansal et al., 2023). A recent method named self-guidance Epstein et al. (2023) proposes to convert the editing signal $g\left(\mathbf{z}_t; t, \mathbf{y}\right)$ into gradients to replace the classifier model in diffusion sampling process. By combining this approach with classifier-free guidance (Ho & Salimans, 2022) as follows, self-guidance achieves high-quality results in editing the location, size and shape of the generate objects.

$$\hat{\epsilon}_t = (1 + s)\epsilon_\theta\left(\mathbf{z}_t; t, \mathbf{y}\right) - s\epsilon_\theta\left(\mathbf{z}_t; t, \varnothing\right) + v\sigma_t\nabla_{\mathbf{z}_t}g\left(\mathbf{z}_t; t, \mathbf{y}\right) \tag{5}$$

where $v$ is an additional guidance weight. In this paper, we aim to design an effective energy function to form a training-free pipeline for text-guided inpainting.

## 2.3 IMAGE INPAINTING

Traditional image inpainting methods (Xu & Sun, 2010; Pérez et al., 2023) leverage heuristic patch-based propagation algorithm to borrow the texture and concept from neighbor of the corrupted regions, which always fail to restore the areas with complex structure or semantics. To remedy this

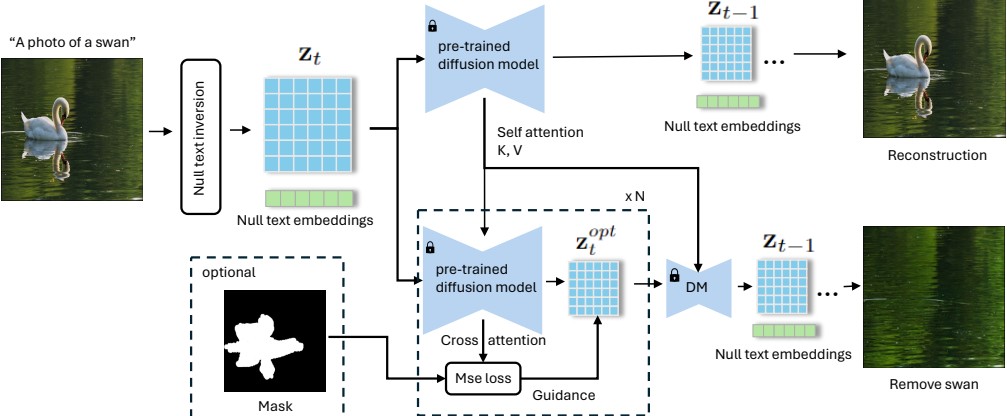

Figure 2: Framework overview. We first invert the input image to obtain the noisy latent $\mathbf{z}_t$ at the t-th time step and optimize a set of null text embedding lists for image reconstruction. Next, we utilize the inversion results to generate the edited image under the guidance of attention. During the editing process, cross-attention mechanisms are employed to obtain optimized image guidance, while self-attention mechanisms are utilized to ensure similarity with the original image.

issue, more recent methods (Guo et al., 2021; Suvorov et al., 2021; Li et al., 2022; Zeng et al., 2021) propose various modifications on CNNs-based image translation networks and use adversarial loss from GANs to improve the consistency of inpainted regions. By synthesizing arbitrary holes on the images, those methods achieve satisfactory performance in general scenes. Some recent attempts(Lugmayr et al., 2022; Xie et al., 2023) have been made to implement diffusion models on image inpainting, demonstrating that diffusion models are capable of produce promising inpainting results. The work most closely related to ours is Inst-Inpaint (Yildirim et al., 2023), which also removes objects from images based on textual descriptions. They follow the data synthesis pipeline from (Brooks et al., 2023) to create large amounts of paired data along with text prompts, which are further used to retrain the standard diffusion to conduct inpainting. In construct to Inst-inpaint, we propose to use the internal attention of the pretrained T2I diffusion models as guidance to produce consistent inpainting results and constrain the other content, which provides a tuning-free pipeline without extra model tuning or data.

# 3 METHOD

## 3.1 OVERVIEW

In this paper, we propose a text-guided image inpainting framework, dubbed ***MagicRemover***, which enables object removal with textual prompts instead of the manual masks from users. The overall framework is shown in Fig. 2. Our method is built upon a T2I diffusion model (*i.e.,* Stable Diffusion) (Rombach et al., 2022b), which consists of an autoencoder and a denoising U-net. Specifically, given an input image $\mathbf{x}_0$ with a textual instruction $\mathbf{y}$ about the objects to be erased (*e.g.,*"A photo of [object class]"), we first adopt Null-text inversion (Mokady et al., 2023) to project them into latent space  for text-image alignment. Then, the projected latent representation $\mathbf{z}_T$ is interacted with text prompt in two parallel U-nets for image reconstruction and inpainting generation, respectively. We propose a novel attention guidance strategy between two branches to generate more consistent filling-in in erasing areas while preserving the other content. Besides, we propose a classifier optimization module to enhance the inpainting capability and stability of the denosing process. With the aforementioned designs, our MagicRemover is capable of removing the objects from textual instruction in a tuning-free manner without extra model training or finetuning. In the following sections, we will introduce the key components of the proposed method, including text-image alignment, attention guidance and classifier optimization.

## 3.2 Text-image alignment

The image inversion module requires users to provide an image and the corresponding text description, which is flexible ranging from a single word of the object to be erased (*e.g.,* [cls]) to a detailed description containing the target object and other information about the image (*e.g.,* a photo of [cls] [scene]). The purpose of text-image alignment is to enable the model to reconstruct the original image based on the given text and initial noise so that it can provide a more accurate spatial response for the target word during the denoising process. We adopt an advanced approach called Null-text inversion (Mokady et al., 2023), which takes the DDIM inversion trajectory $\left\{\mathbf{z}_t^{\text{inv}}\right\}_{t=1}^T$ and the text prompt $\mathbf{y}$ as input, and then optimizes a set of null-text embeddings $\{\varnothing_t\}_{t=1}^T$ according to the following equation:

$$\min_{\varnothing_t} \left\|\mathbf{z}_{t-1}^{\text{inv}} - f_\theta\left(\mathbf{z}_t, t, \mathbf{y}; \varnothing_t\right)\right\|_2^2, \tag{6}$$

where $f_\theta$ represents applying DDIM sampling on the noise predicted by the pre-trained image diffusion model $\epsilon_\theta(\mathbf{x}_t; t, \mathbf{y})$. $t$ indicates the $t - th$ denoising process in totally $T$ steps.

## 3.3 Attention guidance

In our approach, the traditional user-defined masks for indicating erasing areas are discarded, whereas only the text prompts like object classes or referring captions are used as instructions to remove the objects and restore the occluded background. To this end, we leverage the powerful T2I stable diffusion to extract the cross-attention between text prompts and visual latent $\mathbf{z}_t$, and take it as a guidance signal to produce inpainting results. Specifically, for any word with index $k$ in a given text condition $\mathbf{y}$, we can obtain its corresponding cross-attention map $\mathcal{A}_{t,k} \in \mathbb{R}^{H \times W}$ at any timestep $t$. This cross-attention map can be regarded as the spatial response of the image to that specific word. Therefore, optimizing $\mathbf{z}_t$ towards the direction where the response to the $k$-th word is zero naturally leads to the erasure of the object corresponding to the $k$-th word. However, the cross-attention map corresponding to each word not only contains the spatial response to that word but also includes some responses from other objects in the image (see Figure 4). Directly optimizing $z_t$ towards the direction of zero response may cause significant changes in the image background. To overcome this issue, we propose three improvements to construct effective guidance $g(t, k, \lambda)$ from pretrained diffusion models.

**Relaxing constraints.** We optimize $\mathbf{z}_t$ towards a lower response direction rather than an absolute zero matrix. Since the pixel values in the cross-attention map $\mathcal{A}_{t,k}$ represent the degree of relevance to the corresponding word, generally, the values in $\mathcal{A}_{t,k}$ from high to low correspond to the target object, shadows (reflection, etc. if present), and the background. Therefore, we set the optimization target as a matrix filled with pixel values corresponding to the top $20\%$ of the values in the cross-attention map. Empirically, we find that setting the optimization target in this way is sufficient to completely erase the target object. Furthermore, to accommodate users' desires to erase additional associated objects, such as reflections, we provide a hyperparameter $\lambda \in [0, 1]$ to control the degree of object erasure, which is set to 0.8 by default. Therefore, the guidance for erasing the object corresponding to the $k$-th word at timestep $t$ can be defined as follows:

$$g(t, k, \lambda) = \|\mathcal{A}_{t,k} - [\min(\mathcal{A}_{t,k}) + \lambda(\max(\mathcal{A}_{t,k}) - \min(\mathcal{A}_{t,k}))]\mathcal{A}_{t,k}\|_1. \tag{7}$$

For the function of lambda, please refer to Figure 9 in the Appendix.

**Reweight perturbation.** Since the areas with higher relevance to their corresponding word in the cross-attention map have larger values, and those with lower relevance have smaller values, directly multiplying it with the additive perturbation of the classifier guidance can greatly suppress background changes. Therefore, for erasing a single object corresponding to a word, the original attention guidance classifier in Eq. 5 can be rewritten as:

$$\hat{\epsilon}_t^{guid} = (1 + s)\epsilon_\theta\left(\mathbf{z}_t; t, \mathbf{y}\right) - s\epsilon_\theta\left(\mathbf{z}_t; t, \varnothing_t\right) + \mathcal{A}_{t,k} \odot \nabla_{\mathbf{z}_t} g(t, k, \lambda). \tag{8}$$

In some cases, such as removing a specific instance from an image, processing with a single cross-attention map for one word cannot meet the requirements. To further expand the applicability of our

method, we make some modifications to Eq.8. When we want to remove an object described by $n$ words, we change it to the following form:

$$\hat{\epsilon}_t^{guid} = (1 + s)\epsilon_\theta\left(\mathbf{z}_t; t, \mathbf{y}\right) - s\epsilon_\theta\left(\mathbf{z}_t; t, \varnothing_t\right) + (\prod_n \mathcal{A}_{t,k}) \odot (\nabla_{\mathbf{z}_t} \sum_n g(t, k, \lambda)). \tag{9}$$

Additionally, our method supports the use of an extra input mask $M^1$ to label instances in certain extreme cases, such as when there are multiple objects of the same category in an image, making it more difficult to describe the desired instance to erase than obtaining a mask. In this situation, we only need to add an additional term $\nabla_{\mathbf{z}_t}(\hat{\mathbf{z}}_{t-1} - \mathbf{z}_{t-1}^{inv}) \odot (1 - M)$ to Eq.9.

**Self-attention guidance.** Inspired by the consistency in video editing works (Khachatryan et al., 2023; Wang et al., 2023), self-attention plays a vital role in preserving information such as image texture and color. In order to further maintain the similarity between the edited image and the original image, we store the $\mathbf{K_{rec}}$ matrix and $\mathbf{V_{rec}}$ matrix from the reconstruction branch and then modify the self-attention computation formula in the inpainting branch as follows:

$$\text{Attention}(\mathbf{Q}_{edit}, \mathbf{K}_{rec}, \mathbf{V}_{rec}) = \text{softmax}\left(\frac{\mathbf{Q}_{edit}\mathbf{K}_{rec}^T}{\sqrt{d}}\right)\mathbf{V}_{rec}, \tag{10}$$

where, $\mathbf{Q_{edit}}$ matrix comes from the inpainting branch.

### 3.4 CLASSIFIER OPTIMIZATION

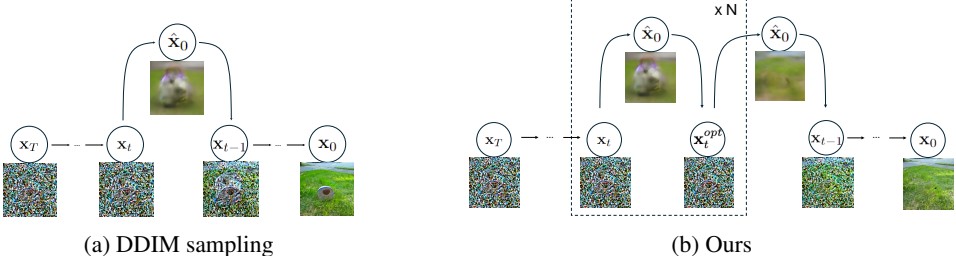

(a) DDIM sampling        (b) Ours

Figure 3: Illustration of Classifier optimization. Our classifier optimization operation can be performed any number of times at any given sampling moment $t$.

In Sec.3.2, we use Null-text inversion (Mokady et al., 2023) to achieve image and text alignment, which requires optimizing Eq.6 for about 100 steps at each sampling moment. Therefore, the more sampling steps we use, the more steps Null-text inversion optimization takes, which is time-consuming. To reduce time while not compromising the editing capability of the algorithm, we propose the classifier optimization operation in this section. Considering that in diffusion-based image editing methods, besides adding perturbations to the predicted noise like classifier guidance sampling, we can also optimize the noisy latent $\mathbf{z}_t$ using gradient descent algorithms through an optimizer (Shi et al., 2023). Given the similarities between these two approaches, we attempt to make some modifications to classifier guidance to achieve the same function as the optimizer, optimizing $\mathbf{z}_t$ without changing $t$. This allows us to increase the optimization steps, reduce the number of sampling steps, and ultimately decrease the time required for Null-text inversion.

Let us reconsider Eq.3, as DDIM can be implemented in the order of first predicting inital latent $\mathbf{z}_0$ from noisy latent $\mathbf{z}_t$ and then adding noise to $\mathbf{z}_{t-1}$. In order to update $\mathbf{z}_t$ without changing $t$, we try to add an optimization step in the denoising process, that is, modifying DDIM to predict $\mathbf{z}_0$ from $\mathbf{z}_t$, and then add noise to $\mathbf{z}_t^{opt}$:

$$\mathbf{z}_t^{opt} = \sqrt{\alpha_t}\left(\frac{\mathbf{z}_t - \sqrt{1 - \alpha_t}\hat{\epsilon}_t^{guid}\left(\mathbf{z}_t\right)}{\sqrt{\alpha_t}}\right) + \sqrt{1 - \alpha_t - \sigma_t^2} \cdot \hat{\epsilon}_t^{guid}\left(\mathbf{z}_t\right) + \sigma_t\epsilon_t \tag{11}$$

To guarantee the resemblance between the edited image and the original image, as well as to regulate the randomness during the sampling process, it is beneficial to set $\sigma_t$ to 0. However, this may result

---

[1]Unless specifically stated, the results presented in this paper are achieved without masks.

in $\mathbf{x}_t^{opt}$ being exactly equal to $\mathbf{x}_t$, which would not serve the purpose of updating $x_t$. Therefore, we replace the noise used in the "predicted $\mathbf{x}_0$" and the "direction pointing to $\mathbf{x}_t$" process with two different noise sources:

$$\mathbf{z}_t^{opt} = \sqrt{\alpha_t} \left( \frac{\mathbf{z}_t - \sqrt{1 - \alpha_t}\hat{\epsilon}_t\left(\mathbf{z}_t\right)}{\sqrt{\alpha_t}} \right) + \sqrt{1 - \alpha_t} \cdot \hat{\epsilon}_t^{guid}\left(\mathbf{z}_t\right) \tag{12}$$

where $\hat{\epsilon}_t = (1 + s)\epsilon_\theta\left(\mathbf{x}_t; t, y\right) - s\epsilon_\theta\left(\mathbf{x}_t; t, \varnothing_t\right)$ refers to the noise predicted through classifier-free guidance approach. The proposed optimization step can be integrated into the denoising process at any given timestep $t$, allowing for the arbitrary updates of $\mathbf{x}_t$ without being restricted by the limitation that the sampling steps should be less than $T$. In our experiments, we found that during the denoising process, using the noise before perturbation $\hat{\epsilon}_t$ for "predicted $\mathbf{x}_0$" and the noise after perturbation $\hat{\epsilon}_t'$ for "direction pointing to $\mathbf{x}_t$" can also improve the editing results, which has been confirmed in the image editing method by Kwon et al. (2022). Detail of our final sampling process can be found in Appendix.

## 4 EXPERIMENTS

### 4.1 IMPLEMENTATION DETAILS

We implement MagicRemover on the publicly available Stable Diffusion 1.5 version (Rombach et al., 2022b). To compare with existing methods, we resize the input image resolution to $512 \times 512$. Since the U-Net used in Stable Diffusion consists of three modules, *i.e.,* encoder, decoder, and bottleneck, with cross-attention maps at four different resolutions at $8 \times 8$, $16 \times 16$, $32 \times 32$, and $64 \times 64$. We uniformly resize all cross-attention maps to the input image's resolution. We then accumulate all cross-attention maps according to the corresponding word and perform normalization to obtain the final cross-attention map used for calculating guidance. Additionally, the diffusion model provides macro information such as layout and content at larger time-steps and generates texture and other details at smaller time-steps. Our pre-trained model has 1000 training steps. We find that incorporating attention guidance at larger time-steps results in a greater erasure strength for objects, but the corresponding cross-attention map is not accurate enough ((as shown in Figure.4)), leading to visually noticeable changes in image tone and background content. Therefore, to achieve a stronger erasure effect while minimizing excessive changes in background content and color tone, we incorporate attention guidance at time-steps $100 < t < t_2$ and add an extra step of classifier optimization for each time-step within the range $t_1 < t < t_2$. By default, $t_1$ and $t_2$ are set to 500 and 800, respectively.

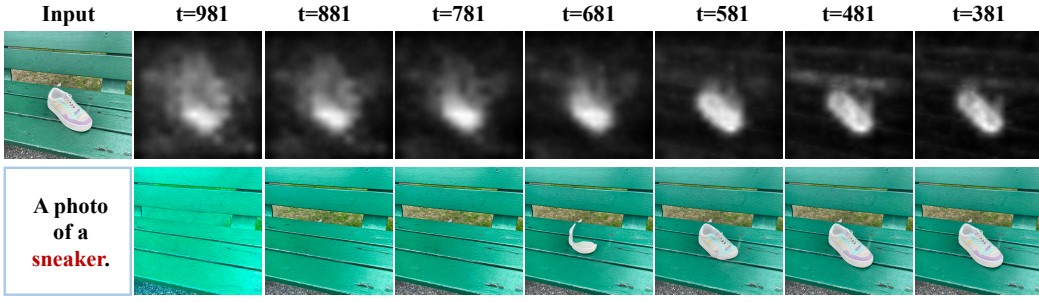

Figure 4: The top row displays the cross-attention maps of the word "sneaker" at various diffusion steps, while the bottom row demonstrates the outcomes obtained by incorporating attention guidance starting from different diffusion steps.

### 4.2 COMPARISON WITH STATE-OF-THE-ART METHODS

We compare our method with three inpainting methods, including two mask-guided approaches (LaMa Suvorov et al. (2021) and SD-Inpaint (Rombach et al., 2022b)) and one text-guided Inst-Inpaint (Yildirim et al., 2023), in terms of quantitative evaluation and user studies.

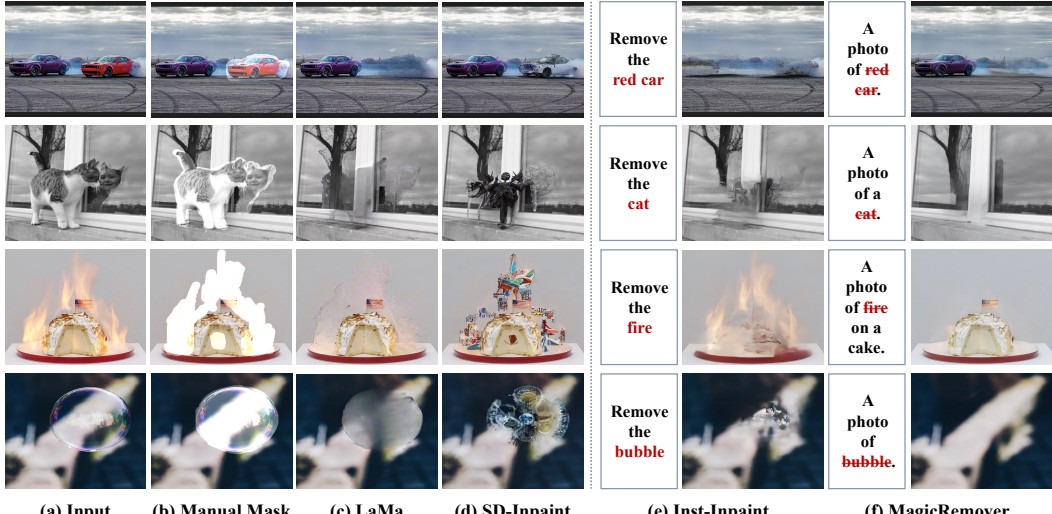

(a) Input  (b) Manual Mask  (c) LaMa  (d) SD-Inpaint  (e) Inst-Inpaint  (f) MagicRemover

Figure 5: Visual comparisons with state-of-the-art methods. From left to right are: (a) input image, (b) manual mask, (c) LaMa (Suvorov et al., 2021), (d) SD-Inpaint (Rombach et al., 2022b), (e) Inst-Inpaint (Yildirim et al., 2023) and (f) MagicRemover. Best viewed by zoom-in on screen.

**Quantitative Results.** Consider the absence of ground truth after object removal, we use Fréchet Inception Distance (FID) (Heusel et al., 2017) to assess the overall quality of the edited images on the COCO 2017 val (Lin et al., 2015). For mask-guided inpainters, we used the ground truth mask from COCO. As shown in Table.1, compared with SD-Inpaint and LaMa, our FID is slightly lower. We believe this is due to the distribution difference caused by the inversion-reconstructed image. Compared with Inst-Inpaint, our method is more accurate in object recognition and does not leave shadows on the target object, resulting in a better FID value. In a vertical comparison, our method includes a mask, which ensures the consistency of the background, thereby obtaining a more favorable FID value.

Table 1: Comparison with sota in terms of FID on COCO 2017 val and the user study result.

| Methods | FID | User Study |
|---|---|---|
| SD-Inpaint | 9.72 | 5% |
| LaMa | 8.44 | 27% |
| MagicRemover w mask | 12.07 | - |
| Inst-Inpaint | 16.15 | 7% |
| MagicRemover w/o mask | 12.78 | 61% |

**User Studies.** Since FID metric can not directly measure the performance of removal operation, We conduct a user study to evaluate the quality of our method. We collect 50 natural images from Internet and DAVIS (Perazzi et al., 2016) and 30 randomly selected ones from COCO 2017val set, resulting in 80 samples in total for user study. For the natural images without mask ground truth, we manually draw a mask for the target object and its effect like shadow or reflection. We ask 30 volunteers to participate in the study and the statistical results are shown in Table.1 demonstrate that our method receives the majority of votes. A subset of the samples in this study are offered in the supplementary materials.

**Qualitative Results.** Fig.5 presents visual comparisons of our method against the other competitors. As seen in Fig.5, existing image inpainting methods cannot preserve background information when removing semi-transparent objects and face challenges in obtaining suitable masks. Although Inst-Inpaint can automatically obtain masks based on text after training, it still fails to acquire appropriate masks when dealing with object removal with soft boundaries. Furthermore, inpainting methods based on generative models can provide text-guided content generation in the mask area. However, they still leave other objects in the mask area and cannot achieve completely clean object removal.

### 4.3 ABLATION STUDY

To validate the effectiveness of the attention strategy proposed in Sec.3, we separately remove relaxing constraints, reweight perturbation, and self-attention guidance, and display the resulting images in Figure.6. As can be seen from Figure.6, Self-attention guidance can greatly maintain the similarity with the original image, while relaxing constraints and reweight perturbation can further ensure the consistency of the image background.

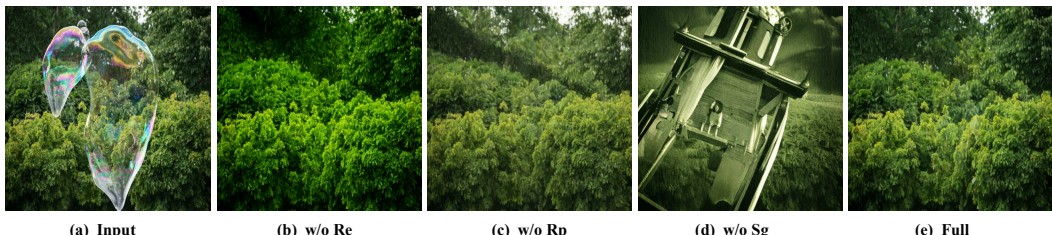

    (a) Input          (b) w/o Re         (c) w/o Rp         (d) w/o Sg         (e) Full

Figure 6: Ablation study on attention guidance: From left to right are: (a) the input images with the corresponding text being "a photo of a bubble,"; (b) to (d) display the output results by removing three components in attention guidance; (e) final output. Here, "Re" stands for relaxing constraints, "Rp" represents reweight perturbation, and "Sg" refers to self-attention guidance.

The effectiveness of the classifier optimization algorithm is demonstrated in Figure 7. (b)-(d) exhibit the results obtained with sampling steps set to 50, 100, and 200, respectively, without using classifier optimization. (e) demonstrates the results achieved with 50 sampling steps and an optimization step added at every timestep during the $t_1 < t < t_2$. It is clear that classifier optimization can enhance the erasure effect of the algorithm. We hypothesize that in the context of a limited number of sampling steps, incorporating classifier optimization into the denoising process can significantly reduce the residual information of the target object, thereby enhancing the algorithm's editing capabilities.

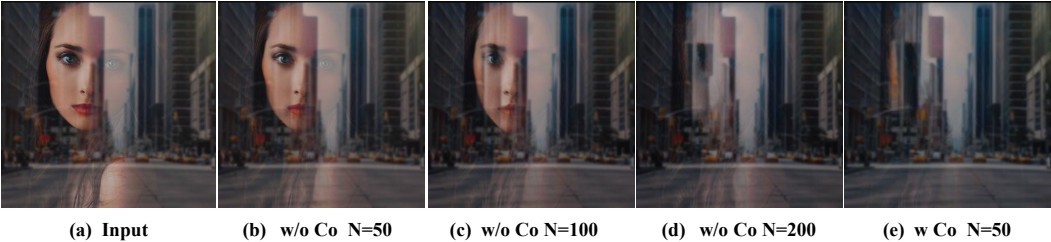

    (a) Input      (b) w/o Co N=50      (c) w/o Co N=100      (d) w/o Co N=200      (e) w Co N=50

Figure 7: Ablation study on classifier optimization: The first column shows the input images, with the corresponding text being "a photo of a woman,". Here, "Co" stands for classifier optimization, and "N" denotes the number of sampling steps.

## 5 CONCLUSION

We propose a tuning-free image inpainting method namely MagicRemover, which harnesses the pretrained diffusion to achieve flexible object removal with only textual instruction. An attention guidance strategy and a classifier optimization algorithm are proposed to constrain the denoising process of diffusion to erase the object and restore the occluded areas. Experimental results in terms of quantitative evaluation and user study demonstrate the superiority of our MagicRemover over state-of-the-art methods.

**Limtations.** Since we obtain the guidance for erasing objects in the latent space of Stable Diffusion, and the encoder used in Stable Diffusion reduces the image resolution to one-eighth of the original, small objects in the image, such as raindrops, cannot be erased. Considering the combination of the encoder and decoder features to acquire guidance might be a solution. We will leave this for future work.

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

# A APPENDIX

## A.1 DETAIL OF OUR SAMPLING ALGORITHM

---

**Algorithm 1** sampling algorithm

---

**Parameter**: Initialize model $\sim \epsilon_\theta$.
Input image $\mathbf{x}_0$
Input caption $\mathbf{y}$
Number of diffusion steps $T$.
Optimization time list opt_t.
OPtimization times $N$.

1: $\mathbf{z}_0 = \mathcal{E}(\mathbf{x}_0)$
2: $\left\{ \mathbf{z}_t^{\text{inv}} \right\}_{t=1}^{T} =$DDIM-inversion$(\mathbf{z}_0)$
3: $\{\varnothing_t\}_{t=1}^{T} =$Null-text-optimization$\left( \left\{ \mathbf{z}_t^{\text{inv}} \right\}_{t=1}^{T}, \mathbf{y} \right)$
4: **for** t in scheduler.timesteps **do**
5:     $\hat{\epsilon}_t = (1 + s)\epsilon_\theta \left( \mathbf{x}_t; t, y \right) - s\epsilon_\theta \left( \mathbf{x}_t; t, \varnothing_t \right)$
6:     $\hat{\epsilon}_t^{guid} = \hat{\epsilon}_t + \sum_m \prod_n \mathcal{A}_{t,k} \nabla_{\mathbf{z}_t} \sum_{m \times n} g(t, k, \lambda)$
7:     $\mathbf{z}_t^{opt} = \mathbf{z}_t$
8:     **if** t in opt_t **then**
9:        **for** i in range(N) **do**
10:           $\mathbf{z}_t^{opt} = \sqrt{\alpha_t} \left( \frac{\mathbf{z}_t - \sqrt{1 - \alpha_t}\hat{\epsilon}_t(\mathbf{z}_t)}{\sqrt{\alpha_t}} \right) + \sqrt{1 - \alpha_t} \cdot \hat{\epsilon}_t^{guid}(\mathbf{z}_t)$
11:        **end for**
12:     **end if**
13:     $\mathbf{z}_{t-1} = \sqrt{\alpha_{t-1}} \left( \frac{\mathbf{z}_t^{opt} - \sqrt{1 - \alpha_t}\hat{\epsilon}_t\left(\mathbf{z}_t^{opt}\right)}{\sqrt{\alpha_t}} \right) + \sqrt{1 - \alpha_{t-1}} \cdot \hat{\epsilon}_t^{guid}\left(\mathbf{z}_t^{opt}\right)$
14: **end for**

---

## A.2 COMPARISON WITH PROMPT-TO-PROMPT

P2P (prompt-to-prompt) (Hertz et al., 2022) proposed three methods to adjust the corresponding attention map during the generation process, namely word swap, adding a new phrase, and attention re-weighting, to achieve the purpose of editing the generated image. A natural question is whether the P2P method, i.e., setting the attention weight of the target object's word to 0 during the reconstruction process or directly deleting the target object's word from the input text, can achieve the function of removing the object. To evaluate this, we conducted a series of experiments and presented partial results in Figure 8. First, we implemented text-image alignment using the method proposed in Sec 3.2 3.2. Then, during the denoising process, we tried using P2P refine and P2P reweighting to erase the target object. P2P refine refers to directly removing the target object's word from the text condition, while P2P reweighting refers to multiplying the cross attention value corresponding to the target object by 0 during the denoising process. As can be seen from the results in Figure 8, the combination of P2P and Null text inversion is unable to remove the target object cleanly.

## A.3 INVESTIGATE THE INFLUENCE OF LAMBDA

In Figure 9, we show the influence of different $\lambda$ values. Since the values in the cross-attention map express the relevance to the text, the lower the lambda value, the cleaner the object will be erased. However, this may also cause some changes in the image's color tone or background. Users may need to adjust the $\lambda$ value to achieve the best results.

## A.4 THE FUNCTION OF MASK

When performing a remove operation on an image, we may encounter objects that are difficult to describe with text or instances that are hard to distinguish through text. For these two situations,

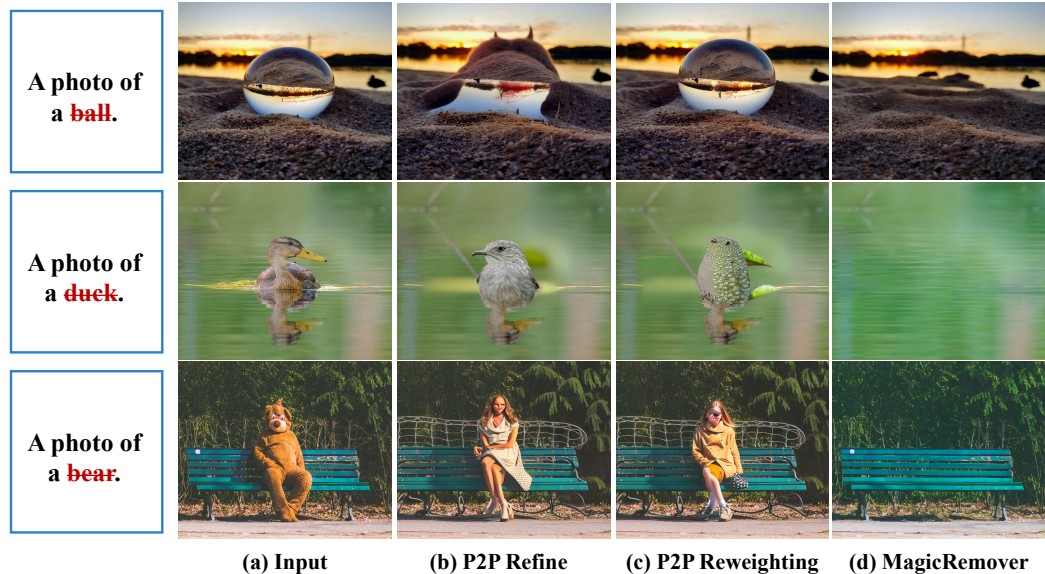

Figure 8: Visual comparisons with Prompt-to-Prompt

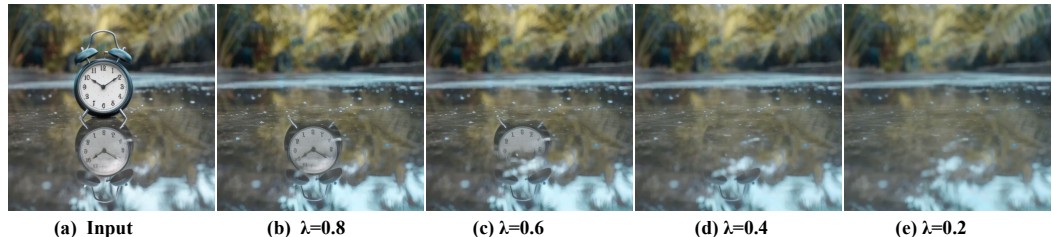

Figure 9: Investigate the influence of lambda.

we need the user to provide an additional mask $M$ to annotate the object to obtain a more accurate result. For the first case, we initially use the mask to cover the object and setting the text condition to "a photo of black stains." Then, we use the method proposed in Sec3.1 to eliminate the 'black stains.' The only change is that since we already have an accurate mask $M$, we can modify Eq.9 to

$$\hat{\epsilon}_t^{guid} = (1+s)\epsilon_\theta\left(\mathbf{z}_t; t, \mathbf{y}\right) - s\epsilon_\theta\left(\mathbf{z}_t; t, \varnothing\right) + M \odot \left(\nabla_{\mathbf{z}_t}\sum_{m \times n} g(t, k, \lambda)\right) \quad (13)$$

to obtain a more precise result, or we can just add an additional term $\nabla_{\mathbf{z}_t}(\hat{\mathbf{z}}_{t-1} - \mathbf{z}_{t-1}^{inv}) \odot (1-M)$ to Eq.9. In our experiments, the results obtained from these two methods did not show any significant differences. For the case of instances that are difficult to distinguish through text, we only need to replace Eq.9 with Eq.13. Figure 10 shows the results for these two situations, respectively.

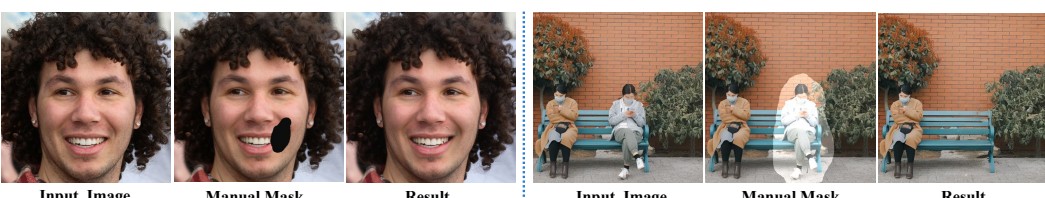

Figure 10: Visual examples in some special cases. In the left image, we use the text "black stains on a photo of a man", and in the right image, we use the text "a photo of two persons."

## A.5 MORE RESULTS

We illustrate more visual comparison results in Fig. 11 and Fig. 12. As we can see, the proposed MagicRemover performs consistently better than other methods on both natural images and coco images. Besides, we list the screenshot of our user study in Fig. 13, more visual results are given in the supplementary zip file.

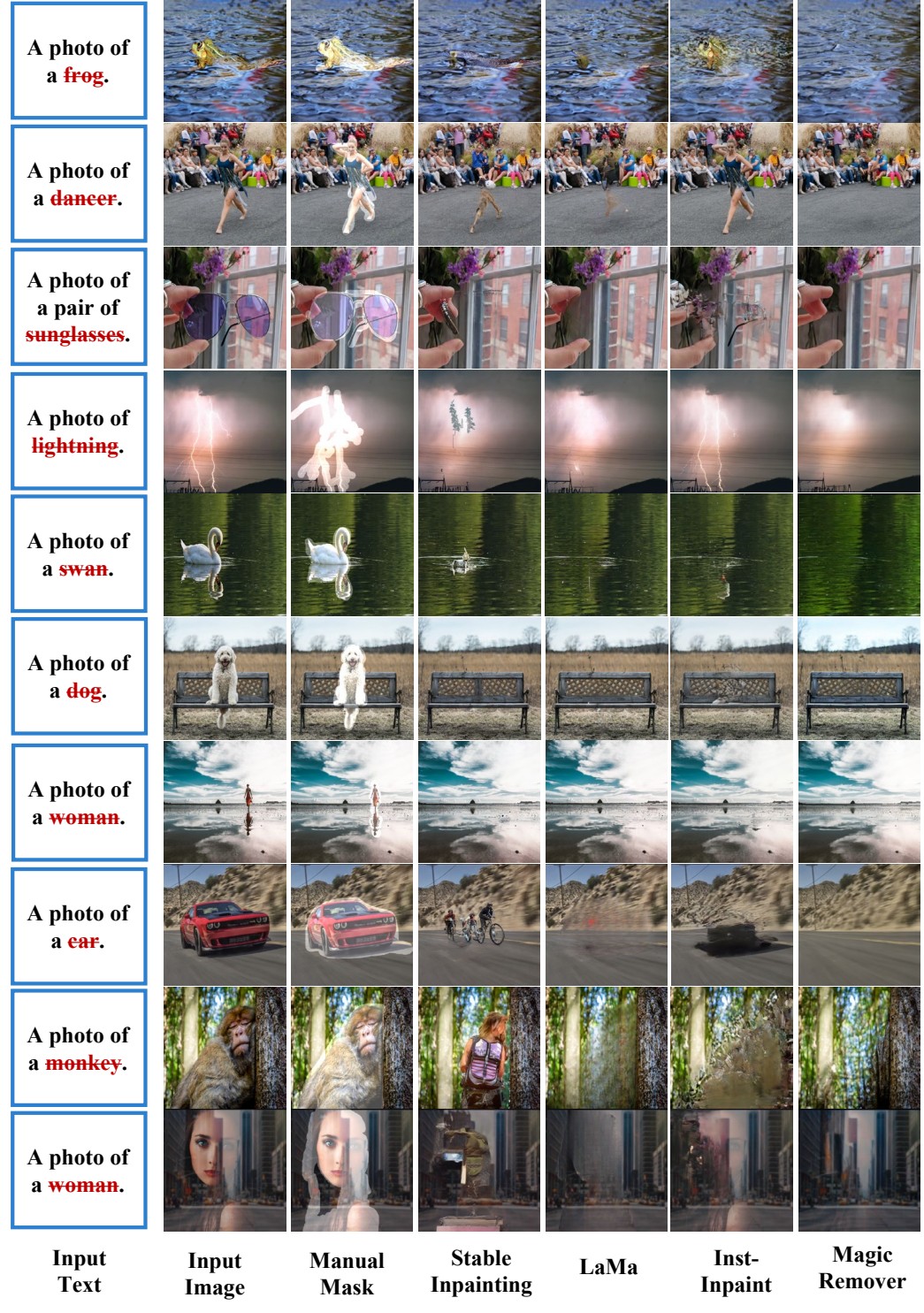

Figure 11: More visual results. Stable inpainting and LaMa utilize manual masks to erase objects. Our method, relies solely on the input text for object erasure. For Inst-Inpaint, we modify the input text from "a photo of..." to "remove the..." in order to obtain results with the object removed.

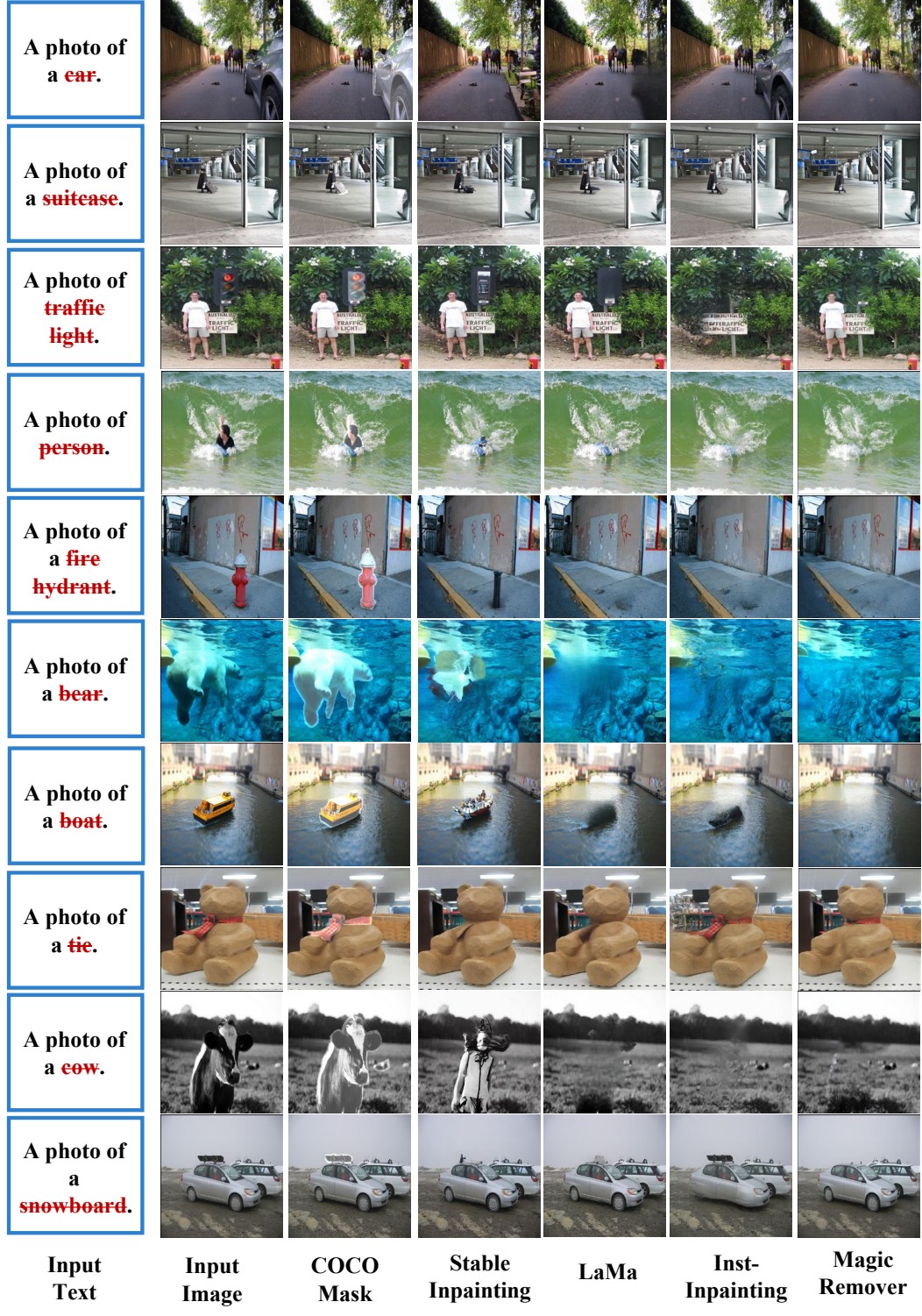

Figure 12: Visual comparisons on coco. Our method relies solely on text-based conditions and adapts the textual input to "remove..." as the condition for inst-inpaint.

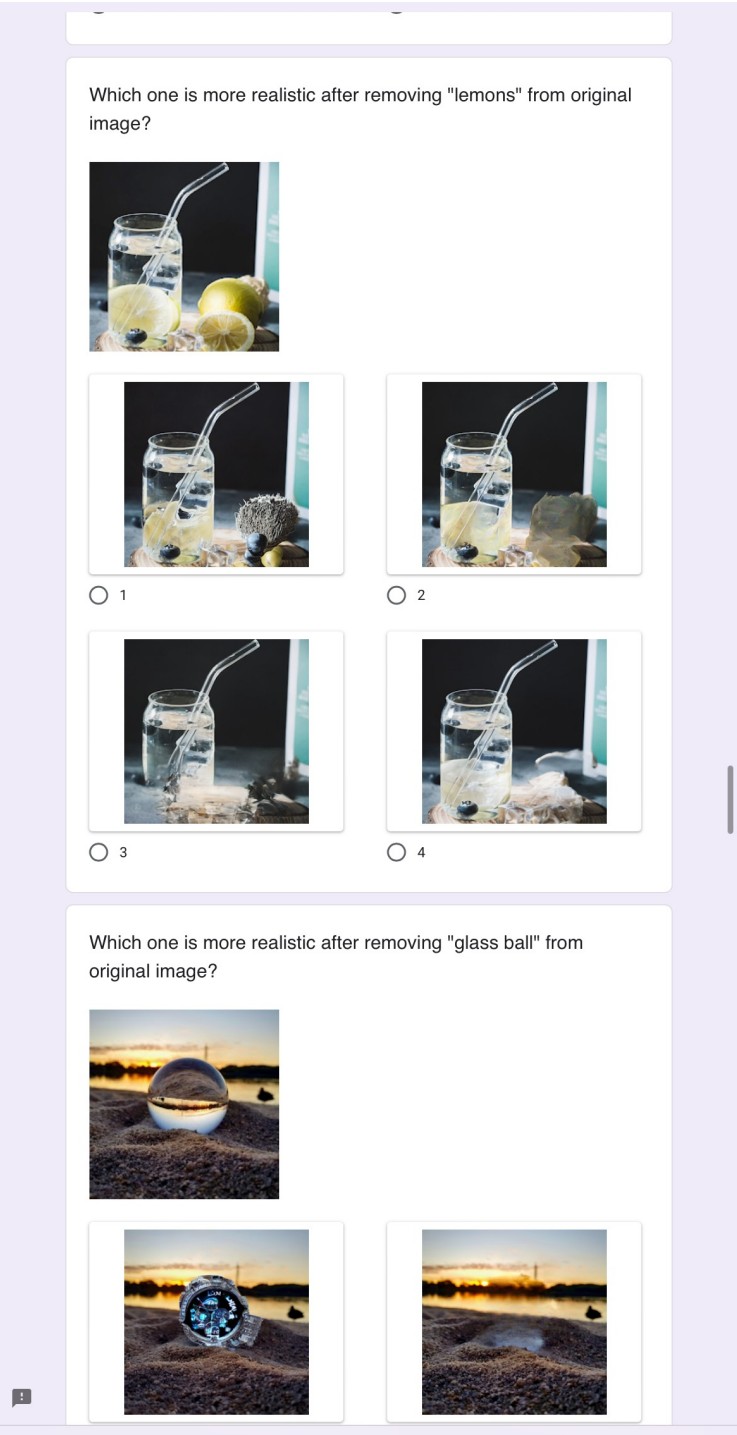

Figure 13: Screenshot of our user study.

