# OpenReview forum: "MagicRemover: Tuning-free Text-guided Image Inpainting with Diffusion Models"
_ICLR.cc/2024/Conference — ICLR 2024 Conference Withdrawn Submission_

### Official Review · Reviewer_L7f6 · 2023-10-25

**Soundness:** 2 fair
**Presentation:** 2 fair
**Contribution:** 2 fair
**Rating:** 5
**Confidence:** 4

**Summary:**

This paper focuses on image inpainting using pre-trained text-to-image diffusion models. Different from previous methods that need to explicitly give the mask, the proposed method takes only textual input from the user to automatically remove the content.

**Strengths:**

The idea of mask-free inpainting is interesting and is especially advantageous in handling removal with soft boundaries or semi-transparent texture, which is challenging to previous inpainting methods.

**Weaknesses:**

The idea of using the cross-attention map for guidance is very similar to previous work P2P [1]. The connection and discrepancy should be discussed in the method.

Missing important citations and discussions on diffusion-based inpainting methods, e.g., DDNM [2] and DPS [3].

The quantitative result in Table 1 is not persuasive enough for the superiority of the proposed method. The robustness of the proposed method is questionable due to the insufficient quantitative experiment.

References:

[1] Hertz et al., Prompt-to-prompt image editing with cross attention control, ICLR 2023

[2] Wang et al., Zero-shot image restoration using denoising diffusion null-space model, ICLR 2023

[3] Chung et al., Diffusion posterior sampling for general noisy inverse problems, ICLR 2023

**Questions:**

I believe this work has advantages in inpainting in subdivided fields like semi-transparent inpainting. Correspondingly, a lot of image restoration problems may be solved using a similar pipeline, e.g., rain removal, fog removal, low light enhancement, etc. In contrast, I think this work is not very advantageous for general inpainting, such as random cropping.

I suggest rethinking the advantages of this method and the topic of this article.

---

### Official Review · Reviewer_zDXi · 2023-10-29

**Soundness:** 3 good
**Presentation:** 3 good
**Contribution:** 2 fair
**Rating:** 5
**Confidence:** 5

**Summary:**

This article presents a tuning-free inpainting method based on stable diffusion. The authors propose an attention-guided strategy and a classifier optimization algorithm, achieving promising results in text-based inpainting.

**Strengths:**

1. The method is distinguished by its training-free nature, resulting in heightened computational efficiency.
2. It demonstrates the capacity to harness the inherent image priors of the diffusion model, thereby yielding favorable outcomes, particularly in the context of object removal.
3. The article exhibits a well-structured presentation with lucid exposition and a straightforward implementation process.

**Weaknesses:**

1. The attention strategy proposed in the article entails multiple hyperparameters. While the authors provide default values, empirical evidence suggests that achieving satisfactory results often necessitates multiple manual adjustments for each image, rendering the process notably inconvenient.
2. The use of attention-based adjustment strategies is a well-established practice in numerous image and video editing tasks, as exemplified by applications such as P2P and FateZero. While the authors have made some adaptations for inpainting tasks, the overarching concept remains largely derivative, lacking substantial innovation.
3. The experimental methodology employed in the study exhibits several shortcomings. The comparison is limited to the ImageNet dataset, overlooking widely used inpainting datasets like FFHQ, CelebA, and Places. Furthermore, the authors rely solely on FID as an evaluation metric, neglecting common indicators such as U-IDS, which compromises the persuasiveness of their findings. Notably, the comparative analysis of text-based inpainting methods is limited, with only one alternative considered, thereby omitting important benchmarks such as NUWA-Inpainting.
4. The ablation study conducted by the authors consists of a single figure, offering limited persuasive value. A more compelling approach would involve a more comprehensive array of case studies and quantitative comparisons.
5. A substantial challenge associated with attention-based methods is their potential to compromise fidelity. This issue is also evident in the method under examination, as evidenced by Figure 9 and Figure 11, where numerous images exhibit noticeable alterations in color and tone.
6. The inference efficiency is not reported. Comparisons to GAN-based and DM-based methods are required.
7. The proposed method is object-oriented. It would be better to provide more cases for background completion.

**Questions:**

1. Is the setting of lambda, t1, and t2 related to the size and shape of objects?
2. Can the proposed method effectively handle situations with occlusion?

---

### Official Review · Reviewer_DDmY · 2023-10-31

**Soundness:** 3 good
**Presentation:** 2 fair
**Contribution:** 3 good
**Rating:** 5
**Confidence:** 3

**Summary:**

This submission proposes a tuning-free diffusion-based method for text-guided image inpainting. An attention guidance strategy is used to guide the erasing of instructed areas and the restoration of occluded content. Moreover, a classifier optimization algorithm is to speed up the denoising procedure. Visual comparisons show that its results are superior to those of baselines.

**Strengths:**

- The proposed techniques are sound and the text-guided image inpainting is interesting.
- The performance is better than the baselines.

**Weaknesses:**

- The framework shown in Figure 2 could better correspond to the proposed key components for a better understanding of this work.
- Most of the proposed key components seem to be borrowed from the existing works. The key contributions could be better clarified.
- The experimental results in the main text are insufficient. The deeper performance analysis of the three key components should be presented, rather than presenting one image sample.
- Considering the performance of the proposed methods is much better than the LaMa and SD-Inpainting from the given image samples, the reason for the lower FID given seems to be unreasonable.
- Is the manual mask in Figure 5 a binary mask for LaMa and SD-Inpaint? Their results are much worse than expected.
- The failure cases are suggested to present.

**Questions:**

Please refer to the above weaknesses.

**Details Of Ethics Concerns:**

It needs a user study to assess the quality of generated images.

---

### Official Review · Reviewer_rxJd · 2023-11-01

**Soundness:** 3 good
**Presentation:** 2 fair
**Contribution:** 2 fair
**Rating:** 5
**Confidence:** 4

**Summary:**

The paper proposes to use the cross attention map in diffusion model to support text-guided object removal.

**Strengths:**

1. Tuning the stable diffusion model to inpainting can cost huge computation resources. The method proposed can save many training time.

2. Insteaf of suppressing the cross attention activation to zero in the corresponding region, the paper proposes to relax the contraint and support shadow removal.

**Weaknesses:**

1. The method need some justification.  In inst-paint, they have paired data, so the model learns to fill in background information.  It is unclear to me why "a lower response direction rather than an absolute zero matrix" can help object removal. Is it possible that it generates a new object rather than filling in background? For instance,  the method can generate a dog inside the cross attention map region to lower the response to the word "horse". Some deep analysis need to be conducted.

2. Some image editing baselines need to be included. The paper is based on null-text and prompt2prompt. So these two methods must be included.  For example, we may recover the embedding with null-text or p2p and use a new word "background" in the corresponding region.

3. Performances. The FID with mask 12.07 is worse than SD-Inpaint and LAMA. Could you provide some analysis?

**Questions:**

as above